# Weisfeiler and Leman Go Infinite:
# Spectral and Combinatorial Pre-Colorings

**Or Feldman**                                                          *orfeldman@campus.technion.ac.il*
*Technion – Israel Institute of Technology, Haifa, Israel*

**Amit Boyarski**                                                          *amitboy@cs.technion.ac.il*
*Technion – Israel Institute of Technology, Haifa, Israel*

**Shai Feldman**                                                  *shai.feldman@campus.technion.ac.il*
*Technion – Israel Institute of Technology, Haifa, Israel*

**Dani Kogan**                                                          *kogan.dani@cs.technion.ac.il*
*Technion – Israel Institute of Technology, Haifa, Israel*

**Avi Mendelson**                                                          *mendlson@technion.ac.il*
*Technion – Israel Institute of Technology, Haifa, Israel*

**Chaim Baskin**                                                          *chaimbaskin@technion.ac.il*
*Technion – Israel Institute of Technology, Haifa, Israel*

**Reviewed on OpenReview:** *https://openreview.net/forum?id=YJDqQSAuB6*

## Abstract

The limit in the expressivity of Message Passing Graph Neural Networks (MPGNNs) has recently led to the development of end-to-end learning GNN architectures. These advanced GNNs usually generalize existing notions in the GNN architecture or suggest new ones that break the limit of the existing, relatively simple MPGNNs. In this paper, we focus on a different solution, the two-phase approach (or pre-coloring), which enables to use of the same simple MPGNNs while improving their expressivity. We prove that using pre-colorings could strictly increase the expressivity of MPGNNs ad infinitum. We also suggest new pre-coloring based on the spectral decomposition of the graph Laplacian and prove that it strictly improves the expressivity of standard MPGNNs. An extensive evaluation of the proposed method with different MPGNN models on various graph classification and node property prediction datasets consistently outperforms previous pre-coloring strategies. The code to reproduce our experiments is available at `https://github.com/TPFI22/Spectral-and-Combinatorial`.

## 1 Introduction

The term Graph Neural Networks (GNNs), as coined by Bronstein et al. (2017), denotes neural networks designed to learn the non-Euclidean structure of graph data. According to Zhou et al. (2020) the two main motivations that led to the modern GNN architectures are the notion of locality and weight sharing as used in CNNs (LeCun et al., 1998), and graph representation learning (Hamilton et al., 2017b). Message Passing Neural Networks (MPNNs or MPGNNs)(Gilmer et al., 2017) are collections of GNNs with common properties. MPGNNs use first order locality by recursively updating the features of each node from its neighborhoods' aggregated features. Then they create a descriptor for the graph by pooling all the node features together. MPGNNs are popular due to their efficiency (Balcilar et al., 2021) and their ability to learn real world graph-structured data (Xu et al., 2019).

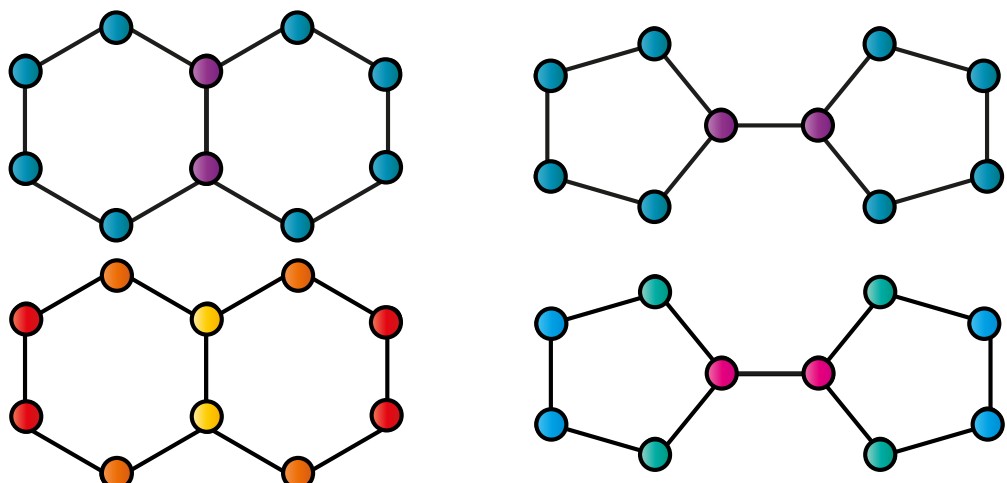

Figure 1: The pair of molecule graphs colored by the degree coloring (upper) and the spectral coloring (lower).

Despite their success, MPGNNs are bounded in their expressive power (i.e., two different graphs may be encoded to the same descriptor by the same MPGNN). It is known that any two graphs that pass the WL test (described in detail in section 2) will be encoded by the same descriptor (Xu et al., 2019). For example, MPGNNs cannot distinguish between the Decalin and Bicyclopentyl molecules graphs (Figure 2) although their graphs are non-isomorphic (Sato, 2020). Attempts have been made to improve the expressive power of MPGNNs by suggesting new and arguably complicated GNN architectures that are not bounded by the WL test, e.g., by using high order networks (Morris et al., 2019), generalizing graphs to simplicial complexes (Bodnar et al., 2021a;b), etc.

In this paper we focus on the two phase approach (or pre-coloring), i.e., generating new features for the nodes before the learning process. This approach is based on the traditional and relatively simple MPGNN architectures and does not require them to be changed at all. We present rigorous proof that the two phase approach can be used to improve the expressiveness of the WL test an infinite number of times in the WL hierarchy (defined in section 2). In addition, we propose an expressive permutation equivariant pre-coloring based on the spectral decomposition of the graph Laplacian that is also efficient to compute, explainable, and generates constant size features with respect to the graph size. Figure 1 demonstrates the improvement in expressivity that our suggested pre-coloring adds compared to the relatively simple degree coloring, on the Decalin and Bicyclopentyl molecules graphs. Using our proposed pre-coloring, one can easily differentiate between the two graphs, even though standard MPGNNs and the WL test cannot distinguish between them.

**Contributions.**

- We prove that the expressive power of WL can be improved ad infinitum by a sequence of equivariant pre-colorings and that each of the latter can be computed in polynomial time. Thus, the upper bound of the existing MPGNNs can be improved accordingly. This contribution serves as motivation for the two phase approach.

- We suggest expressive and informative pre-coloring based on the spectral decomposition of the graph Laplacian, and explicitly prove that it improves the expressivity of the WL test, and MPGNNs accordingly.

- We perform extensive experiments showing that this simple extension consistently improves the performance of various MPGNNs on different benchmarks, compared to previous works.

## 2   Preliminaries

**Graph isomorphism.**   An undirected graph of size $N$ is a pair $\mathcal{G} = (V, E)$ where $V = \{v_1...v_N\}$ is a set of vertices and $E$ is a set of edges. Each edge is a set of two vertices from $V$. We say that two graphs $\mathcal{G}_1 = (V_1, E_1)$ and $\mathcal{G}_2 = (V_2, E_2)$ are isomorphic if there exists bijection $\sigma : V_1 \rightarrow V_2$ s.t. $\{v_i, v_j\} \in E_1 \iff \{\sigma(v_i), \sigma(v_j)\} \in E_2$. There is no known polynomial time algorithm for determining isomorphism between any arbitrary pair of graphs. Nevertheless, there are some classes of graphs (trees, planar) between which isomorphism can be determined using the polynomial time algorithm $k$-WL test (Kiefer et al., 2019; Immerman & Lander, 1990). Moreover, Grohe (2012) showed that for almost any class of graphs, the $k$-WL test can determine isomorphism.

**Graph coloring.**   Graph coloring is a mapping from a vertex and its graph to a label (color), from a known set of labels. We say that coloring $C$ *refines* coloring $D$ if for any two graphs $\mathcal{G}_1$ and $\mathcal{G}_2$, and for any two vertices $v_1 \in V_1$, $v_2 \in V_2$ s.t. $C(v_1) = C(v_2)$, then $D(v_1) = D(v_2)$. A column vector $x$ can be used to represent the node colors of a coloring $C$ where $C(V_i) = x_i$, for a specific graph. We say that $C$ is *permutation equivariant* if for any two graphs with adjacency matrices $\mathbf{A}$ and $\mathbf{B}$, and column vectors $x, y$ representing $C$ on $\mathbf{A}$ and $\mathbf{B}$ respectively s.t. $\mathbf{A} = \mathbf{P}\mathbf{B}\mathbf{P}^T$ where $\mathbf{P}$ is a permutation matrix, then $x = \mathbf{P}y$. Ideally, we would like to find the following permutation equivariant coloring: for each $v_1 \in V_1, v_2 \in V_2$, $C(v_1) = C(v_2) \iff$ there exists isomorphism $\sigma : \mathcal{G}_1 \rightarrow \mathcal{G}_2$ s.t., $\sigma(v_1) = v_2$.

**k-WL test.**   The Weisfeiler-Leman (WL) (Weisfeiler & Leman, 1968) test of isomorphism is an algorithm for testing a necessary but insufficient condition for graph isomorphism. Two graphs that do not pass the test are necessarily non-isomorphic. First, the algorithm assigns to each node the same color using the constant coloring $C_{WL}^0(v) = \text{CONST}$. Then the algorithm continues with iterations. At each iteration $i$, each node receives its neighbors' colors, and together with its color, it generates a new color for the next iteration, i.e., $C_{WL}^i(v) = (C_{WL}^{i-1}(v), \{\{C_{WL}^{i-1}(x) | x \in \mathcal{N}(v)\}\})$, where '$\{\{\}\}$' denotes a multi-set, and $\mathcal{N}(v)$ denotes the set of neighbors of $v$. This process continues until convergence whereupon the colors are collected into a histogram. If the two graphs have different histograms, they failed the test and are called distinguishable. If after the convergence, the two histograms are the same, the graphs did not fail. Having thus passed the test, they are called indistinguishable. It was proved by Bevilacqua et al. (2022) that $C_{WL}^{i+1}$ always *refines* $C_{WL}^i$. The WL test can be extended to k-tuple coloring instead of vertex (1-tuple) coloring (the full definition of the algorithm can be found in the appendix). This extension is called the $k$-WL test. It was proved by Cai et al. (1992) that any pair of graphs that are indistinguishable by $k + 1$-WL are also indistinguishable by $k$-WL. Moreover, for any $k \geq 2$, there exists a pair of graphs s.t. they are distinguishable by $k + 1$-WL but indistinguishable by $k$-WL, i.e., $k + 1$-WL is *strictly more expressive* than $k$-WL, for $k \geq 2$. This hierarchy of expressiveness is called the *WL hierarchy*. The diagonal $k$-WL coloring on the graph vertices is defined to be $\Delta(k\text{-}WL)(v) = C_{k-WL}(v, ..., v)$ where $C_{k-WL}$ is the coloring after the $k$-WL converges. It was proved by Rattan & Seppelt (2021) that $\Delta(k + 1\text{-}WL)$ *refines* $\Delta(k\text{-}WL)$.

**Message Passing Graph Neural Networks.**   MPGNNs are a specific type of GNNs. MPGNNs work in layers; each layer $l$ has its own Multi Layer Perceptron $(\text{MLP})_l$ and iterates over the nodes in the graph. For each node, its neighbors' features are aggregated together with its own features using some aggregation operation. The result of the aggregation is then used as input to the MLP of the current layer, and the output is the node's new features. To create a descriptor of the graph, the node features of each layer are aggregated separately, and the results are combined together. In other words, the node features of vertex $v$ after $l$ layers are $h_v^{(l)} = \text{MLP}_{(l)}(\text{UPDATE}(h_v^{(l-1)}, \text{AGGREGATE}(\{\{h_x^{(l-1)} | x \in \mathcal{N}(v)\}\})))$ where the graph descriptor is $h_{\mathcal{G}} = \text{COMBINE}(\{\text{AGGREGATE}(\{\{h_x^{(l)} | x \in \mathcal{N}(v)\}\}) | l \in \texttt{layers}\})$. It was proved by Xu et al. (2019) that the expressive power of MPGNNs is bounded by the expressive power of 1-WL, i.e., for any two graphs $\mathcal{G}_1$ and $\mathcal{G}_2$ s.t. $\mathcal{G}_1$ and $\mathcal{G}_2$ are indistinguishable by 1-WL, their descriptors created by any MPGNN will be equal. Moreover, it was proved that MPGNNs whose node features are aggregated using summation are strictly more expressive than MPGNNs that use other popular operations such as MAX and MEAN. The MPGNN based on summation is called Graph Isomorphism Network (GIN), and it has also been shown to produce state-of-the-art results in addition to the theoretical superiority in expressive power.

**Graph Laplacian** Let $\mathcal{G} = (V, E)$ be an undirected graph with an adjacency matrix denoted by $\mathbf{A}$. Given a function $\mathbf{x} \in \mathbb{R}^{|V|}$ on the vertices, the Dirichlet energy of the function $\mathbf{x}$ on the graph is defined to be

$$\mathbf{x}^\top \mathbf{L} \mathbf{x} = \sum_{(v,u) \in E} \mathbf{A}(v, u) \left( x(v) - x(u) \right)^2. \tag{1}$$

The matrix $\mathbf{L}$ is the *(combinatorial) graph Laplacian*, and is given by $\mathbf{L} = \mathbf{D} - \mathbf{A}$, where $\mathbf{D}$ is the *degree matrix*, i.e., diagonal matrix where $\mathbf{D}(v, v) = |\mathcal{N}(v)|$. $\mathbf{L}$ is symmetric and positive semi-definite and, therefore, admits a spectral decomposition $\mathbf{L} = \mathbf{\Phi} \mathbf{\Lambda} \mathbf{\Phi}^\top$. Since the sum of each row in $\mathbf{L}$ is 0, $\lambda_1 = 0$ is always an eigenvalue of $\mathbf{L}$. The eigenpairs $(\phi_i, \lambda_i)$ can be thought of as the graph analogue of 'harmonic' and 'frequency'. The graph Laplacian is the discrete generalization of the Laplace-Beltrami operator and hence it has similar properties to it.

The spectrum of the graph Laplacian holds structural information about it. For example, the multiplicity of the zero eigenvalue represents the number of connected components in the graph. Another example is the second eigenvalue (counting multiple eigenvalues separately) that measures the connectivity of the graph (Spielman, 2009). We say that a pair of graphs are cospectral or cospectral with respect to the Laplacian if their spectra of the Laplacians are equal.

**Heat kernel.** The heat equation associated with the Laplacian is given by:

$$\frac{\partial H_t}{\partial t} = -L H_t \tag{2}$$

where $H_t$ is the heat kernel and $t$ is time. The heat kernel describes the process of heat diffusion on the graph through time. Each column of the heat kernel matrix represents a different diffusion process on the graph. When each column $i$ is initialized with a value of 1 on the $i$-th entry and the rest of the entries are zeros the solution of the heat equation is:

$$H_t = e^{-tL} \tag{3}$$

Then the heat kernel at time $t$ can be computed by exponentiating the Laplacian eigenspectrum:

$$H_t = \Sigma_{i=1}^{|V|} e^{-\lambda_i t} \phi_i \phi_i^T \tag{4}$$

where $\lambda_i$ is the $i$-th eigenvalue of the graph Laplacian and $\phi_i$ is its corresponding eigenvector. The element at the index $(u, v)$ can also be computed by:

$$H_t(u, v) = \Sigma_{i=1}^{|V|} e^{-\lambda_i t} \phi_i(u) \phi_i(v) \tag{5}$$

$H_t(u, v)$ is the amount of heat transferred from node $u$ to node $v$ until time $t$, from the beginning of a diffusion process where all the nodes had 0 heat and u had exactly 1. When the observed point in time $t$ tends to zero, the kernel is affected mostly by the local structures of the graphs. When the observed time point is relatively large, the global structure of the graphs becomes the dominant structure.

## 3 Expressive power of 1-WL with pre-colorings

In section 2 we noted that the expressive power of 1-WL is limited. In particular, it is strictly limited by the expressive power of $k$-WL, for any $k > 2$. In this section, we present theoretical support for the two phase approach by showing that the expressive power of the standard 1-WL algorithm can be improved up to anywhere in the *WL hierarchy* when using pre-colorings.

If we pre-color 1-WL with a coloring $C$, we mark the new algorithm as 1-$C$WL.

**Theorem 1.** *Let $R_1, R_2$ be two colorings s.t. $R_2$ refines $R_1$ and $R_2$ is permutation equivariant. Accordingly, 1-$R_2$WL is at least as expressive as 1-$R_1$WL.*

Proof outline (the full proof can be found in the appendix):

1. We show that for any pair of isomorphic graphs, their histogram of 1-$R_2$WL is the same when using the permutation equivariant property of the coloring.

2. We show that two graphs distinguishable by 1-$R_1$WL are also distinguishable by 1-$R_2$WL. This is a corollary of the color refinement property of the 1-WL iterations.

For $R_1$ and $R_2$ that satisfy Theorem 1, it is enough to find a single pair of graphs that are indistinguishable by 1-$R_1$WL but distinguishable by 1-$R_2$WL to prove strictness in expressive power.

**Theorem 2.** *Let $\mathcal{G}_1,\mathcal{G}_2$ be any two graphs. Their $\Delta$(k-WL) histograms are equal $\iff$ their $C_{k-WL}$ histograms are equal.*

Proof outline (the full proof can be found in the appendix):

1. We prove the first direction by running $k$-WL for extra $k-1$ iterations after it converged. Following the structure of the $k$-WL, we show that the coloring of the graph is effectively "folded" onto the diagonal. Since these iterations happen after convergence, they do not change the final coloring, which means that the full graph histogram is encoded onto the diagonal tuples.

2. We prove the second direction by the fact that in the initialization of $k$-WL, if a tuple is colored with the same color as a diagonal tuple, then it is necessarily a diagonal tuple. It, therefore, retains this property throughout the iterations.

**Theorem 3.** *For any $k \geq 2$, 1-$\Delta$(k+1-WL)WL is strictly more expressive than 1-$\Delta$(k-WL)WL.*

A corollary of Theorem 2 and Theorem 3 is that 1-WL can become as powerful as $k$-WL for any $k$ when using $\Delta(k\text{-}WL)$ as a pre-coloring. This means that the expressive power of MPGNNs, which is provably bounded by the expressive power of 1-WL, can be improved ad infinitum in the WL hierarchy using the right permutation equivariant pre-coloring as a pre-process before the MPGNN learning phase.

Not every permutation equivariant coloring $C$ makes 1-$C$WL strictly more expressive than 1-WL.

**Example 3.1.** *If $D(u) = |\mathcal{N}(u)|$, i.e., the degree coloring, then 1-DWL is equal to 1-WL in terms of expressive power.*

## 4 Spectral pre-coloring

**Spectral WL**   We propose an expressive pre-coloring based on the spectrum of the graph Laplacian, which can be used to color the nodes instead of the constant coloring of the 1-WL algorithm. We will call this variant the *spectral WL algorithm*. To calculate the pre-coloring, we first compute $m$ heat kernel matrices for evenly spaced points in time on the logarithmic scale. Then for each node $u$, we give the following color: $(H_{t_1}(u,u), ..., H_{t_m}(u,u))$. Finally, we choose a constant amount of quantiles $r$ from the row of $u$ (ignoring the element on the diagonal) and append them in ascending order, e.g., $((q_{1_u}^{t_1}...q_{r_u}^{t_1}), ...(q_{1_u}^{t_m}...q_{r_u}^{t_m}))$, to the existing color of the node. The spectral coloring with a simple setting of $m = 1$, $t = 1$, and no quantiles is sufficient to compute the ideal equivariant coloring of the molecules graphs mentioned in section 1.

**Theorem 4.** *The spectral pre-coloring is permutation equivariant.*

Many spectral based methods generate node features that depend on the spectral decomposition representation of the graph Laplacian (Srinivasan & Ribeiro, 2020; Dwivedi & Bresson, 2020; Kreuzer et al., 2021). In section 5 we demonstrate the flaws of this property. By Theorem 4 we get that the spectral pre-coloring that we suggest does not depend on the representation of the spectral decomposition of the Laplacian and, therefore, does not suffer from these flaws.

**Theorem 5.** *Spectral WL is strictly more expressive than 1-WL.*

Proof outline (the full proof can be found in the appendix):

1. We prove that *spectral WL* is as expressive as 1-WL using Theorem 4 and Theorem 1 and the fact that any coloring *refines* the constant coloring.

2. We show a concrete example where *spectral WL* can distinguish between a pair of non-isomorphic graphs and 1-WL does not.

Although *spectral WL* is strictly more expressive than 1-WL, the exact expressive power of spectral-based approaches is currently unknown and it is stated as an open question by Fürer (2010). However, Fürer (2010) also showed that specific spectral methods based on the adjacency matrix are bounded in their expressive power by 2-WL. Rattan & Seppelt (2021) reached the same conclusion for the graph Laplacian and other matrices that are computed by the adjacency matrix.

**Spectral features for GNNs.** This pre-coloring can be used to create initial node features for MPGNNs as a pre-process before the learning phase. Instead of applying the coloring, we can append it to the existing node features of any graph. As hinted by Theorem 5, in the results of the experiments we will see that it is enough to add a relatively small feature vector, e.g., with 10 entries, to achieve great expressivity even for real world graphs with hundreds and thousands of nodes. One can, however, *refine* the pre-processing by adding more quantiles and time samples. The features that we added to each node have the desirable property of being explainable, and they have the following meaning: For node $u$, the feature at entry $i \leq m$ is the amount of heat left at $u$ at time $t_i$ from the beginning of a diffusion process where all the nodes had 0 heat and $u$ had exactly 1. The features at entries $i > m$ represent the distribution of the heat diffusion through time on the other nodes.

**Scaling for large graphs.** For enormous graphs, calculating all the eigenvalues and eigenvectors is impractical. For large and small values of $t$, one can approximate $\mathbf{H}_t$ via spectral or spatial techniques. For large values of $t$ and $\lambda$, $e^{-t\lambda}$ become negligible, hence it is enough to rely on the $k$ smallest eigenvalues. For small values of $t$, $\mathbf{H}_t$ can be computed iteratively using explicit/implicit Euler iterations. Alternatively, we can use the model order reduction (MOR) technique to obtain approximate dynamics. The technique has been used successfully for computing isometry invariant descriptors for shape analysis (Bähr et al., 2018). To that end, given the discretized heat equation

$$\dot{\mathbf{x}} + \mathbf{L}\mathbf{x} = \mathbf{0}, \tag{6}$$

we use the eigendecomposition $\mathbf{L} = \mathbf{\Phi}\mathbf{\Lambda}\mathbf{\Phi}^\top$ to obtain

$$\mathbf{\Phi}^\top\dot{\mathbf{x}} + \mathbf{\Lambda}\mathbf{\Phi}^\top\mathbf{x} = \mathbf{0}. \tag{7}$$

Since the dynamics are governed by the smaller eigenvalues of $\mathbf{L}$, we can truncate the eigendecomposition to obtain a lower-dimensional approximation of the dynamics. Denoting $\mathbf{w}_k = \mathbf{\Phi}_k^\top\mathbf{x}$, with $\mathbf{\Phi}_k$ being the first (smallest) $k$ eigenvectors of $\mathbf{L}$, we compute the approximated heat kernel at time $t$ by integrating

$$\dot{\mathbf{w}}_k + \mathbf{\Lambda}\mathbf{w}_k = \mathbf{0} \tag{8}$$

up until the relevant time. This is a simple PDE with a diagonal matrix $\mathbf{\Lambda}$ that can easily be solved via the unconditionally stable implicit Euler method.

Despite the fact that we thus obtain only an approximation of the heat kernel, it is important to remember that our task is not to solve the heat equation but rather to provide useful spectral features for graph learning tasks. In the related problem of non-rigid shape retrieval, descriptors obtained via the approximated dynamics provide better retrieval results compared to descriptors based on the full dynamics (Bähr et al., 2018). An approximation analysis of the suggested scaling technique can be found in the appendix.

## 5 Experimental study on synthetic benchmarks

To demonstrate the improvement in expressivity that the spectral features add, we built two benchmarks, each of which is based on a single pair of graphs. The first pair of graphs is the Decalin and Bicyclopentyl molecule graphs that have the same 1-WL histogram but their spectrum is different (Sato, 2020). The second pair of graphs is presented in Figure 3, and the graphs are distinguishable by 1-WL but cospectral

with respect to the Laplacian. For each benchmark, we created 1000 examples by adding or removing a single edge at random from the original graphs and reordering their node indices randomly. For each benchmark, we split all the instances into training and test sets with a ratio of 9:1. The goal of a classifier for the benchmark is: Given a graph from the test set, identify the original graph from which it was perturbed. We trained GIN (Xu et al., 2019), GCN (Kipf & Welling, 2017), GraphSAGE (Hamilton et al., 2017a) and GAT (Veličković et al., 2018) and their appropriate Spectral Pre-processed (SP) classifiers with the same settings of five message passing layers, a hidden dimension of 64, a learning rate of 0.01 and spectral features from 10 points in time using only the maximum quantile, for 100 epochs. We repeated the experiment 100 times and report the average accuracy and standard deviation of each classifier. We used **bold** to denote significantly high accuracy for each type of MPGNN.

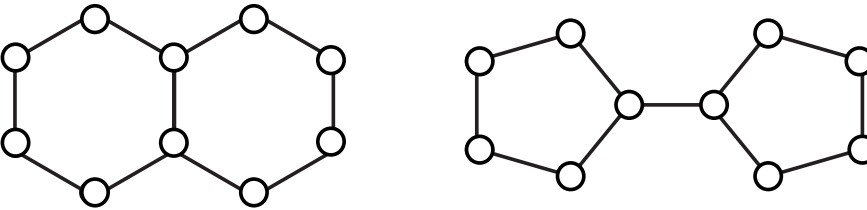

Figure 2: First pair of the original graphs.The graphs are 1-WL indistinguishable but not cospectral.

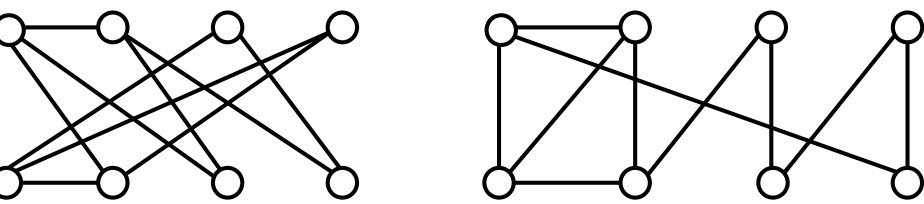

Figure 3: Second pair of the original graphs. The graphs are 1-WL distinguishable but cospectral with respect to the Laplacian.

Table 1: Experimental study results

| GNN / Test set | 1-WL indistinguishable | Cospectral |
|---|---|---|
| GIN | 64.1±4.0 | 93.3±2.5 |
| SP-GIN | **99.0±5.2** | 93.3±4.5 |
| GCN | 51.5±4.0 | 73.5±16.7 |
| SP-GCN | **98.1±6.5** | 92.1±5.5 |
| GAT | 50.2±0.9 | 49.0±0.8 |
| SP-GAT | **97.7±11.8** | **77.6±16.0** |
| GraphSAGE | 49.8±0.9 | 49.4±0.9 |
| SP-GraphSAGE | **95.1±12.1** | **91.6±6.8** |

From the results in Table 1, for the 1-WL indistinguishable pair of graphs, the MPGNNs struggle to identify the source of each graph. This is probably because 1-WL cannot differentiate between the sources. The spectral features help them to overcome this issue easily. GIN, which has the most expressive aggregation

operation among all the MPGNNs, achieves great accuracy on the cospectral graphs; the other uniformly colored MPGNNs, however, do not. These results make sense since cospectral graphs have common structural properties. In Figure 4 and Figure 5 we can see the spectral coloring of the cospectral graphs introduced by the spectral pre-prossessing – nodes with the same color have the same spectral features. In Figure 4 the pre-processing does not use any quantiles and in Figure 5 the pre-processing uses only the maximum quantile. We can see that not only do both colorings *strictly refine* the constant coloring, but that the coloring that uses the maximum quantile *strictly refines* the one that does not.

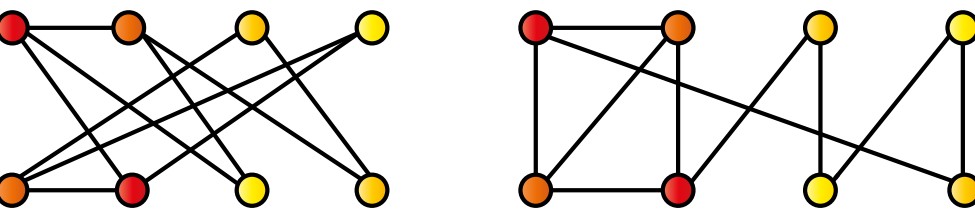

Figure 4: Cospectral graph coloring based only on the diagonal of the heat kernel.

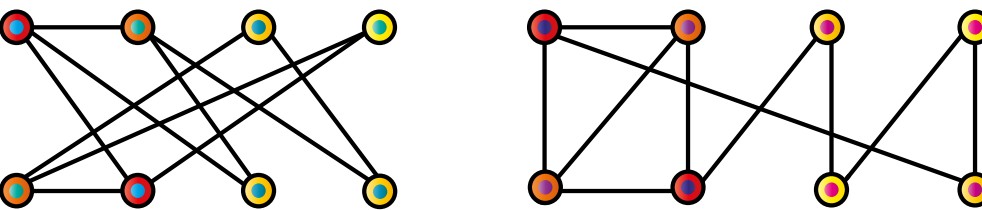

Figure 5: Cospectral graph coloring based on the diagonal of the heat kernel and the maximum quantile.

In addition to the theoretical justification in Theorem 1 we conducted a synthetic experiment that demonstrates the importance of the permutation equivariant property of pre-colorings. Non permutation equivariant pre-colorings such as random node initialization (Sato et al., 2021; Abboud et al., 2021) are able to distinguish between two 1-WL indistinguishable graphs, but at the cost of generating different embeddings for isomorphic graphs, which makes them seemingly expressive. Hence, we trained GIN on a dataset containing permuted examples of two 1-WL indistinguishable graphs pre-colored with random node initialization. Given a permuted graph, the purpose of the classifier is to identify to which of the two 1-WL indistinguishable graphs the permuted graph is isomorphic. We used GIN with a hidden dimension of 64, a learning rate of 0.01, 50 epochs, and repeated each training-testing session 10 times. Since GIN generated different embeddings for isomorphic graphs, it achieved poor performance of 50% average accuracy. When we switched the random node initialization pre-coloring with our suggested spectral pre-coloring the average accuracy increased to 100%.

## 6 Evaluation on real benchmarks

We evaluate our pre-processing method on two graph learning tasks: graph classification and node classification. For each task, we used four types of MPGNNs (GIN, GCN, GAT, and GraphSAGE) from the Pytorch Geometric framework (Fey & Lenssen, 2019) to compare the standard use of the network to our SP method. Statistics regarding all of the benchmarks we used can be found in the appendix.

### 6.1 Graph classification

For this part of the evaluation, we used standard benchmarks and settings that were suggested by Xu et al. (2019) and followed by many other works (Kolouri et al., 2021; Cai et al., 2021; Maron et al., 2019; Bouritsas et al., 2022; Bodnar et al., 2021a;b). We compared our method to previously suggested techniques of equivariant pre-coloring that generate constant size features. The setting includes eight graph classification

benchmarks: five social network datasets (COLLAB, IMDB-BINARY, IMDB-MULTI, REDDITBINARY, and REDDIT-MULTI5K), and three molecule datasets (MUTAG, PTC, NCI1) (Yanardag & Vishwanathan, 2015). The task of the benchmarks here is to achieve the highest average validation accuracy with 10-fold cross-validation. We used GNNs with five layers where in each layer's MLP a single hidden layer was used. We used concatenation to create the final graph descriptor and a linear layer to create the final output. We fine-tuned the dropout of the linear layer to be one of {0,0.5}. For the molecule datasets, we fine-tuned the hidden dimension of all the MLPs to be one of {16,32}, while for the social network benchmarks we consistently used a hidden dimension of size 64. The number of epochs that achieved the best cross-validation accuracy, averaged over the 10 folds, was selected. We examined 700 epochs for each configuration. For a fair comparison, we did not tune any of the SP parameters and used constant configuration of 10 time samples on the logarithmic scale from $t_1 = 10^{-2}$ to $t_{10} = 10^2$, and no quantiles. We report the average validation accuracy and standard deviation over 10 folds. The results for the molecules datasets are located in Table 2, and the results for the social networks datasets are located in Table 3.

Table 2: Graph classification results – Molecules
**Bold** accuracy denotes the best coloring for a specific type of MPGNN on a specific benchmark and **_underlined italic bold_** accuracy denotes the best coloring for any MPGNN and any coloring on a specific benchmark.

| Method | MUTAG | PTC | NCI1 |
|---|---|---|---|
| GIN | 89.3±5.2 | 65.6±6.5 | **82.1±1.4** |
| LDP-GIN[1] | 87.7±9.2 | 69.0±4.1 | 79.6±1.6 |
| 3SI-GIN[2] | 89.3±7.4 | 69.0±8.1 | 80.6±1.8 |
| SP-GIN (ours) | **90.9±9.4** | **69.3±12.0** | 81.5±1.7 |
| GCN | 79.3±9.1 | 69.3±10.0 | 81.4±1.3 |
| LDP-GCN[1] | 88.2±7.7 | **70.2±7.8** | 79.2±1.5 |
| 3SI-GCN[2] | 89.2±6.0 | 68.7±8.5 | 80.0±1.3 |
| SP-GCN (ours) | **_93.0±5.8_** | 69.4±9.3 | **81.7±1.4** |
| GAT | 81.9±7.4 | 69.0±8.8 | **81.2±1.4** |
| LDP-GAT[1] | 88.7±10 | 68.7±5.9 | 78.8±1.8 |
| 3SI-GAT[2] | 90.4±6.2 | 68.4±9.0 | 79.9±2.2 |
| SP-GAT (ours) | **91.4±6.8** | **69.0±8.7** | **81.2±1.9** |
| GraphSAGE | 84.0±1.1 | 68.1±9.1 | 81.9±16 |
| LDP-GraphSAGE[1] | 90.3±9.9 | 68.7±10.0 | 80.1±1.7 |
| 3SI-GraphSAGE[2] | 89.8±5.6 | 68.1±7.4 | 80.6±1.2 |
| SP-GraphSAGE (ours) | **92.4±5.5** | **_70.8±6.0_** | **_82.9±1.4_** |

[1] Local Degree Profile (Cai & Wang, 2018), [2] 3 Subgraph Isomorphism counting (Bouritsas et al., 2022).

On most of the benchmarks, both molecules and social networks, the SP-MPGNNs performed better than other pre-colored MPGNNs.

## 6.2 Node classification

We used node classification as an additional method of evaluation and used four node classification benchmarks for this task: three citation network datasets (Cora, CiteSeer, and PubMed) (Yang et al., 2016) and

Table 3: Graph classification results – Social networks
**Bold** accuracy denotes the best coloring for a specific type of MPGNN on a specific benchmark and **_underlined italic bold_** accuracy denotes the best coloring for any MPGNN and any coloring on a specific benchmark.

| Method | COLLAB | IMDB-B | IMDB-M | REDDIT-B | REDDIT-M |
|---|---|---|---|---|---|
| GIN | 70.4±2.4 | 72.7±4.9 | **_51.3±3.6_** | 79.6±3.9 | 53.7±2.1 |
| LDP-GIN[1] | 73.8±1.4 | 72.6±4.3 | 49.7±4.2 | 86.1±2.3 | 55.7±2.0 |
| 3SI-GIN[2] | 73.1±2.3 | 72.1±4.6 | 49.2±4.1 | — | — |
| SP-GIN (ours) | **76.4±2.3** | **73.4±4.5** | 51.0±4.2 | **86.9±2.0** | **56.9±1.6** |
| GCN | 76.6±2.5 | 65.4±4.5 | 41.5±2.8 | 90.9±1.6 | 57.1±1.9 |
| LDP-GCN[1] | 74.8±2.5 | 68.6±3.1 | 44.4±2.6 | **_92.1±0.7_** | 57.5±1.5 |
| 3SI-GCN[2] | 75.5±1.8 | 64.6±3.7 | 42.5±3.4 | — | — |
| SP-GCN (ours) | **76.7±1.9** | **_73.6±4.5_** | **49.8±4.6** | 91.9±2.5 | **57.7±1.3** |
| GAT | 36.5±7.8 | 52.6±3.1 | 36.4±2.5 | 72.5±5.9 | 24.3±6.3 |
| LDP-GAT[1] | **74.2±1.9** | 69.8±5.4 | 44.6±2.0 | 90.6±2.0 | 56.8±1.7 |
| 3SI-GAT[2] | 73.9±1.9 | 65.8±6.0 | 43.5±2.8 | — | — |
| SP-GAT (ours) | 73.7±2.5 | **73.0±3.7** | **50.4±3.6** | **_92.1±1.5_** | **57.8±1.3** |
| GraphSAGE | 36.6±7.9 | 53.4±2.2 | 35.7±4.1 | 74.6±3.4 | 35.7±2.3 |
| LDP-GraphSAGE[1] | 76.8±1.2 | 71±4.3 | 44.7±2.8 | 91.4±1.7 | 57.1±1.1 |
| 3SI-GraphSAGE[2] | **_77.6±1.5_** | 68.5±4.0 | 43.8±3.1 | — | — |
| SP-GraphSAGE (ours) | 74.4±2.8 | **73.2±3.3** | **50.2±4.0** | **91.8±1.8** | **_57.9±1.7_** |

[1] Local Degree Profile (Cai & Wang, 2018), [2] 3 Subgraph Isomorphism counting (Bouritsas et al., 2022).

a biochemistry dataset (PPI) (Zitnik & Leskovec, 2017). The task of the benchmarks here is to achieve the highest average test accuracy upon 100 random initializations of the GNNs. For the citation networks, only the number of message passing layers, the hidden dimension of the MLPs, and the number of training epochs were fine-tuned, using the validation set. The number of the layers was one of {2, 3, 4}, the hidden dimension was one of {128, 256, 384, 512} and each model was trained for at most 200 epochs. Specifically for PPI, there were two layers, the hidden dimension was 512 and the models were trained for 800 epochs. The spectral pre-process was calibrated exactly as in the synthetic experiment. We repeated each training-testing session 100 times and report in Table 4 the average accuracy and standard deviation of the test set. We used the standard train-test split supplied by Pytorch Geometric.

Table 4: Node classification results
**Bold** denotes significantly high accuracy for each type of MPGNN .

| Method | CiteSeer | Cora | PubMed | PPI |
|---|---|---|---|---|
| GIN | 71.9±0.6 | 81.8±0.5 | 79.6±0.5 | 91.1±0.2 |
| SP-GIN (ours) | 71.3±0.6 | 81.9±1.8 | 78.8±0.7 | 91.4±0.2 |
| GCN | 63.5±4.4 | 78.1±2.6 | 80.4±0.5 | 88.8±0.1 |
| SP-GCN (ours) | **72.1±0.8** | **82.3±1.4** | 80.8±0.4 | **89.2±0.1** |
| GAT | 64.1±4.5 | 81.6±1.0 | 79.9±1.3 | 79.6±0.2 |
| SP-GAT (ours) | **72.3±1.5** | 79.2±1.9 | 80.4±0.7 | **80.7±0.3** |
| GraphSAGE | 72.8±0.6 | 82.9±0.9 | 80.2±0.6 | 95.8±0.1 |
| SP-GraphSAGE (ours) | 72.9±0.6 | 81.9±2.3 | 80.8±0.5 | **96.0±0.1** |

Even though each node in the benchmark contains a feature vector with hundreds of entries, appending to it a relatively small number of spectral features usually improved the accuracy of the MPGNNs. It can be explained by the fact that the spectral features also contain global information about the graph and the node's position. This information cannot be learned using a small amount of message passing iterations. It is known that deeper MPGNNs (many message passing iterations), however, suffer from two phenomena called Over-squashing (Alon & Yahav, 2021; Topping et al., 2022) and Over-smoothing (Wu et al., 2020; Li et al., 2018; Chen et al., 2020) that are also make it difficult for them to learn global graph information.

## 6.3 Ablation study

Alongside the benchmark testing we conducted, we also wanted to examine the effect of the parameters of our spectral pre-processing method, including the number of points in time to sample, the range of the sample, and the quantiles to use. For this experiment, we chose one benchmark with initial features (NCI1) and one benchmark with none (COLLAB). We again used the four MPGNNs. Each was trained using a hidden dimension of 64 with five message passing layers, for 700 epochs. We report the average and standard deviation of the test accuracy on 10 different folds of the datasets. The configuration setting is reported as '(start of the sampling range in powers of 10, end of the sampling range in powers of 10, number of samples, used quantiles)'.'MMM' in the quantiles entry denotes the use of the max, min and median quantiles. The full results can be found in the appendix.

For the NCI1 benchmark, the range and the quantiles amount with which we chose to train the MPGNNs achieve the best results when the number of features is limited to 10. For the NCI1 and COLLAB benchmarks, we can see that the performance of SP-MPGNNs can be improved even further by choosing spectral features with more than 10 entries.

# 7    Related works

This section surveys works related to our research. We split the section into paragraphs by the method used to improve the expressive power of MPGNNs – spectral methods and methods for generalizing the message passing scheme.

**Use of spectral decomposition in GNNs.**    Work has been done to improve the expressive power of GNNs, with some studies adopting the spectral based approaches. An example of such an approach is the SAN architecture (Kreuzer et al., 2021). First, SAN finds the spectral decomposition of the graph Laplacian using the k-th smallest eigenvalues and their appropriate eigenvectors. Then it encodes them into node features using a transformer with self-attention. Another example is the DGN architecture (Beani et al., 2021), which uses the eigendecomposition of the Laplacian to calculate the derivative or direction between the nodes. The directions are then encoded as node features for the GNN. Both SAN and DGN have been shown to produce state-of-the-art results on real world benchmarks. These methods, however, allow different descriptors for isomorphic graphs since they are dependent on the eigendecomposition representation.

**Breaking the limits of MPGNNs**    Several other works tried to break the limit of expressivity of MPGNNs. Some introduce new sophisticated GNN architectures that are based on interesting concepts from various fields of research. These concepts usually generalize the GNNs' message passing scheme, which makes the new architectures more expressive but less efficient. The first attempt to extend the expressive power of GNNs was the K-Dimensional Graph Neural Network (Morris et al., 2019). These networks generalize MPGNNs in the same way the $k$-WL test generalizes the WL test. Hence, their expressive power is naturally better, upper bounded by the $k$-WL instead of the WL. Unfortunately, these networks require tremendous memory and computation time as K increases, similar to the $k$-WL test.

The Simplicial Isomorphism Network (SIN) (Bodnar et al., 2021b) is another approach that extends the expressive power of GNNs. This method treats graphs as a general algebraic object called a simplicial complex and performs the message passing between every two neighbors in the simplicial complex instead of the adjacent vertices. Bodnar et al. proved that SIN is strictly more powerful than the WL test and at least as powerful as the 3-WL test. The Cell Isomorphism Network (CIN) (Bodnar et al., 2021a) is yet another architecture based on an algebraic object. This object is called a regular cell complex and it generalizes the simplicial complex. In their paper, the researchers use the definition of 'cell complex adjacencies' to define the new scheme of message passing. Similar to SIN, it was proved that CIN is strictly more powerful than the WL test and at least as powerful as the 3-WL test. They also present great results on learning tasks for molecular problems.

# 8    Discussion

In this work we demonstrated how one can strictly improve the expressive power of the WL test an infinite number of times in the WL hierarchy using the diagonal coloring of the $k$-WL algorithm, and simultaneously improve the upper bound for MPGNNs, without any change in their architecture. We also proposed spectral pre-processing for MPGNNs that is based on the diagonal and quantiles of the heat kernel matrix. From the results of the graph classification and node classification benchmarks, we conclude that our method of pre-processing improves the performance of MPGNNs on real world graph-structured data, compared to previous works. For example, the classification accuracy of GIN, the most expressive MPGNN, on social networks graphs, improved by 3.5%, and when using GAT the improvement is much more significant and stands at 17%, when comparing it to the uniform pre-coloring. Moreover, our suggested pre-coloring consistently outperformed previously suggested pre-colorings on wide range of real world graph classification benchmarks. We also saw that the spectral pre-coloring is a possible research direction for dealing with the Over-squashing problem.

## 9  Acknowledgement

This research was partially supported by the Technion Hiroshi Fujiwara Cyber Security Research Center, the Israel National Cyber Directorate and the Pazy foundation.

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

# A  Proofs

In the following proofs we assume the definition of the $k$-WL as defined by Morris et al. (2019). $C_{k-WL}^0$ is defined to be equal between any two tuples of vertices from $\mathcal{G}_1$ and $\mathcal{G}_2$, if and only if the two subgraphs of $\mathcal{G}_1$ and $\mathcal{G}_2$ comprising all the vertcies in each tuple are isomorphic. We first define the multiset for iteration $i$ at index $j$ to be $c_{k-WL}^{i,j}(v_1, ...v_k) = \{\{C_{k-WL}^{i-1}(v_1, ...v_{j-1}, w, v_{j+1}, ...v_k) | w \in V\}\}$. Finally, we define the $k$-WL coloring at iteration $i$ on tuple $s$ to be $C_{k-WL}^i(s) = (c_{k-WL}^{i,1}(s), ...c_{k-WL}^{i,K}(s))$.

## A.1  Theorem 1 proof

*Proof.* Let $\mathcal{G}_1$ and $\mathcal{G}_2$ be two isomorphic graphs where $\sigma : V_1 \to V_2$ is the isomorphism. We will prove by induction that after $n$ message passing iterations of 1-WL initilized with permutation equivariant coloring, $R_2$, the coloring of every pair $v \in V_1$ and $u \in V_2$ s.t. $\sigma(v) = u$ is the same.

**Base (n=0):** $R_2$ is permutation equivariant and hence by its definition $R_2(v) = R_2(u)$ for each $v \in V_1$ and $u \in V_2$ s.t. $\sigma(v) = u$.

**Step:** From the induction assumption we know that every two nodes $v \in V_1$ and $u = \sigma(v) \in V_2$ have the same color after $n$ message passing iterations of 1-WL. For each such $v$ and $u$ we will look at the coloring after the $n + 1$ iteration of 1-WL. These are equal to $(C_{1-R_2WL}^n(v), \{\{C_{1-R_2WL}^n(x) | x \in \mathcal{N}(v)\}\})$ and $(C_{1-R_2WL}^n(u), \{\{C_{1-R_2WL}^n(x) | x \in \mathcal{N}(u)\}\})$, respectively. $C_{1-R_2WL}^n(u)$ and $C_{1-R_2WL}^n(v)$ are equal from the induction assumption. $\sigma$ is an isomorphism and hence $x \in \mathcal{N}(v) \iff \sigma(x) \in N(u)$ and, therefore, $\{\{C_{1-R_2WL}^n(x) | x \in \mathcal{N}(v)\}\}$ and $\{\{C_{1-R_2WL}^n(x) | x \in \mathcal{N}(u)\}\}$ are equal. Since $\sigma$ is a bijection, we get that the coloring histogram of $\mathcal{G}_1$ and $\mathcal{G}_2$ is the same for each $n$.

Let $\mathcal{G}_1$ and $\mathcal{G}_2$ be any two graphs and let $R_1, R_2$ be two initial colorings for 1-WL s.t. $R_2$ *refines* $R_1$. We will prove by induction that for each $v \in V_1$ and $u \in V_2$, s.t. $C_{1-R_2WL}^n(v) = C_{1-R_2WL}^n(u)$, $u, v$ also satisfy $C_{1-R_1WL}^n(v) = C_{1-R_1WL}^n(u)$ for any number $n$ of 1-WL message passing iterations. Therefore, if $C_{1-R_1WL}^n(v) \neq C_{1-R_1WL}^n(u)$ then $C_{1-R_2WL}^n(v) \neq C_{1-R_2WL}^n(u)$.

**Base (n=0):** For any $v \in V_1$ and $u \in V_2$, if $C_{1-R_2WL}^0(v) = C_{1-R_2WL}^0(u)$ then $C_{1-R_1WL}^0(v) = C_{1-R_1WL}^0(u)$ since $R_2$ *refines* $R_1$.

**Step:** Let $v \in V_1$ and $u \in V_2$ be any two vertices s.t. $C_{1-R_2WL}^{n+1}(v) = C_{1-R_2WL}^{n+1}(u)$. Their coloring in the $n + 1$ iteration is equal to $(C_{1-R_2WL}^n(v), \{\{C_{1-R_2WL}^n(x) | x \in \mathcal{N}(v)\}\})$ and $(C_{1-R_2WL}^n(u), \{\{C_{1-R_2WL}^n(x) | x \in \mathcal{N}(u)\}\})$, respectively. From the induction assumption we find that $C_{1-R_1WL}^n(v) = C_{1-R_1WL}^n(u)$. In addition, we know that the two multisets in the second part of the tuples are equal, this means that there exists an injective mapping $\mu : \mathcal{N}(u) \to \mathcal{N}(v)$ s.t. $C_{1-R_2WL}^n(x) = C_{1-R_2WL}^n(\mu(x))$ and hence by the induction assumption $\{\{C_{1-R_1WL}^n(x) | x \in \mathcal{N}(v)\}\} = \{\{C_{1-R_1WL}^n(x) | x \in \mathcal{N}(u)\}\}$ and therefore $C_{1-R_1WL}^{n+1}(v) = C_{1-R_1WL}^{n+1}(u)$.

If $\mathcal{G}_1$ and $\mathcal{G}_2$ are 1-$R_1$WL distinguishable they have different 1-$R_1$WL histograms after some iteration $n$. Hence, an injective mapping $\mu : V_1 \to V_2$ s.t. $C_{1-R_1WL}(x) = C_{1-R_1WL}(\mu(x))$ for any $x \in V_1$, does not exist. From the claim proved by induction, an injective mapping $\mu : V_1 \to V_2$ s.t. $C_{1-R_2WL}(x) = C_{1-R_2WL}(\mu(x))$ for any $x \in V_1$ does not exist. Therefore $\mathcal{G}_1$ and $\mathcal{G}_2$ have different $1 - R_2$WL histograms and are distinguishable by 1-$R_2$WL.

$\square$

## A.2  Theorem 2 proof

1. *Proof.* Given $\mathcal{G}_1$ and $\mathcal{G}_2$ s.t. $\{\{C_{K-WL}(v_1, ...v_k) | v_1, ...v_k \in V_1\}\} = \{\{C_{k-WL}(v_1, ...v_k) | v_1, ...v_k \in V_2\}\}$, we will prove that $\{\{\Delta(k\text{-}WL)(v) | v \in V_1\}\} = \{\{\Delta(k\text{-}WL)(v) | v \in V_2\}\}$. From the initialization of $k$-WL we know that the color of each tuple of the form $(v, .., v)$ is equal only to other tuples of this form since they are the only ones that represents a graph with a single vertex. Hence, for any $v \in V_1$ and $u_1, u_2, ...u_k \in V_2$ if $C_{k-WL}(v, ..., v) = C_{k-WL}(u_1, ..., u_k)$; then necessarily $u_1 = u_2 = ... = u_k$. Since any $v \in V_1$ is injectively mapped to $u \in V_2$ with the same diagonal coloring, we get that $\{\{\Delta(k\text{-}WL)(v) | v \in V_1\}\} = \{\{\Delta(k\text{-}WL)(v) | v \in V_2\}\}$.

In the second part of the proof we are given $\mathcal{G}_1$ and $\mathcal{G}_2$ s.t. $\{\{\Delta(k\text{-}WL)(v)|v \in V_1\}\} = \{\{\Delta(k\text{-}WL)(v)|v \in V_2\}\}$ and we need to prove that $\{\{C_{k-WL}(x_1,...,x_k)|x_1,...,x_k \in V_1\}\} = \{\{C_{k-WL}(x_1,...,x_k)|x_1,...,x_k \in V_2\}\}$.

From the definition of $k$-WL, it immediately derived that if for any $(x_1, x_i, x_{i+1}, ...x_k) \in V_1^{k-i+2}$ there exists $(y_1, y_i, y_{i+1}...y_k) \in V_2^{k-i+2}$ s.t. $K\text{-WL}(x_1, x_1, ...x_1, x_i, x_{i+1}...x_k) = K - WL(y_1, y_1, ...y_1, y_i, y_{i+1}...y_k)$, then for any $(x_1, x_{i-1}, x_i, ...x_k) \in V_1^{k-i+3}$ there exists $(y_1, y_1, ...y_1, y_{i-1}, y_i...y_k) \in V_2^{k-i+3}$ s.t. $K\text{-}WL(x_1, x_1, ...x_1, x_{i-1}, x_i...x_k) = K\text{-}WL(y_1, y_1, ...y_1, y_{i-1}, y_i...y_k)$. Since we know that this condition holds for $i = k + 1$, i.e. constant tuples, it also accrues for $i = k, k - 2, ...2$. Since the condition holds for $i = 2$, $\{\{C_{k-WL}(x_1, ..., x_k)|x_1, ..., x_k \in V_1\}\} = \{\{C_{k-WL}(x_1, ..., x_k)|x_1, ..., x_k \in V_2\}\}$ also holds. $\qquad\square$

## A.3 Theorem 3 proof

*Proof.* From Theorem 1 it immediately is derived that $1\text{-}\Delta(k+1\text{-}WL)$WL is as expressive at least as $1\text{-}\Delta(k\text{-}WL)$WL. To show that this inequality is strict, we will find a pair of graphs for each $K \geq 2$ s.t. they are indistinguishable by $1\text{-}\Delta(k\text{-}WL)$WL but distinguishable by $1\text{-}\Delta(k+1\text{-}WL)$WL. For any $K \geq 2$ we know there exists $\mathcal{G}_1$ and $\mathcal{G}_2$ s.t. they are distinguishable by $k+1$-WL and indistinguishable by $k$-WL. From Theorem 2 we know that this pair of graphs is also distinguishable by the $1\text{-}\Delta(k+1\text{-}WL)$WL algorithm. We also know from Theorem 2 that the $\Delta(k\text{-}WL)$ histograms of the graphs are equal. We will prove that the $1\text{-}\Delta(k\text{-}WL)$WL histograms of the graphs are also equal by showing that the message passing iterations of 1-WL does not change the nodes' colors except for the marking/representation of the colors, i.e., the message passing iterations of the 1-WL does not add any new information to the coloring. After a single iteration of $1\text{-}\Delta(k\text{-}WL)$WL, the new coloring of any vertex $v$ is $(\Delta(k\text{-}WL)(v), \{\{\Delta(k\text{-}WL)(u)|u \in \mathcal{N}(v)\}\})$, i.e., the new information added to the coloring is the coloring histogram of the neighbors. We will show that this information can be derived from $\Delta(k\text{-}WL)(v)$ for any $v$. From the initialization of $k$-WL, we can find in any color of a tuple $(v, v, ...u)$ such that $u \in \mathcal{N}(v)$ since their representing graphs are isomorphic and different from the representing graphs for $(v, v, ...x)$ where $x \notin \mathcal{N}(v)$. In this way, we can find any color of a tuple $(v, u, ...u)$ s.t. $u \in \mathcal{N}(v)$. Again from the initialization of $k$-WL we can find the color of any $(u, u, ...u) = \Delta(k\text{-}WL)(u)$ s.t. $u \in \mathcal{N}(v)$.

Since the coloring of $1\text{-}\Delta(k\text{-}WL)$WL does not change in any iteration and because the coloring histograms are equal from the beginning, $1\text{-}\Delta(k\text{-}WL)$WL cannot distinguish between the pair of graphs. $\qquad\square$

## A.4 Example 1 proof

*Proof.* We will prove that $C_{1-WL}^1 \equiv D$, i.e., the coloring generated after a single iteration of 1-WL initialized with constant coloring equals $D$. For any vertex $v$, it is colored with the following coloring: $(C_{1-WL}^0(v), \{\{C_{1-WL}^0(x)|x \in \mathcal{N}(v)\}\}) = (CONST, \{\{CONST, CONST, ...CONST\}\})$ where the multiset size is equal to the size of $\mathcal{N}(v)$. Hence $C_{1-WL}^1 \equiv D$.

$\qquad\square$

## A.5 Theorem 4 proof

*Proof.* Even though the eigenvalues and eigenvectors of the Laplacian are being used to compute the heat kernel, it does not depend on their representation (e.g. $\phi$ or $-\phi$). This is because the eigenspectrum is being used here to compute the matrix exponentiation in Equation 3 which is well defined. Now we prove that the spectral pre-coloring is indeed permutation equivariant. Let $\mathcal{G}_1$ and $\mathcal{G}_2$ be two graphs and let $\mathbf{A}$ and $\mathbf{B}$ be their associated adjacency matrices s.t., $\mathbf{A} = \mathbf{PBP}^T$ for some permutation matrix $\mathbf{P}$. In order to prove that the spectral pre-coloring is permutation equivariant it is sufficient to prove that $H_t^{\mathbf{A}} = \mathbf{P}H_t^{\mathbf{B}}\mathbf{P}^T$ for every $t$. We know that $L_\mathbf{A} = \mathbf{PL_BP}^T$. Hence, $\mathbf{L_A}$ and $\mathbf{L_B}$ have the same set of eigenvalues, and if $\phi_i$ is an eigenvector of $\mathbf{L_B}$ then $\mathbf{P}\phi_i$ is an eigenvector of $\mathbf{L_A}$. Then:

$$H_t^A = \Sigma_{i=1}^{|V|}e^{-\lambda_i t}\mathbf{P}\phi_i\phi_i^T\mathbf{P}^T = \mathbf{P}(\Sigma_{i=1}^{|V|}e^{-\lambda_i t}\phi_i\phi_i^T)\mathbf{P}^T = \mathbf{P}H_t^B\mathbf{P^T} \qquad (9)$$

$\qquad\square$

| Coloring Histogram | | | | | | |
|---|---|---|---|---|---|---|
| Color
Graph | 0.1914 | 0.1929 | 0.2891 | 0.291 | 0.3078 | 0.3098 |
| $\mathcal{G}_1$ | 2 | 0 | 4 | 0 | 4 | 0 |
| $\mathcal{G}_2$ | 0 | 2 | 0 | 4 | 0 | 4 |

Table 5: Coloring histograms after initialization of *Spectral WL*

### A.6 Theorem 5 proof

*Proof.* From Theorem 1 it is immediately derived that *Spectral WL* is as expressive at least as 1-WL since the spectral pre-coloring is permutation equivariant and any coloring *refines* the constant coloring. We will show that there exist two graphs that are indistinguishable by 1-WL but distinguishable by *Spectral WL* and hence *Spectral WL* is strictly more expressive than 1-WL.

Let $\mathcal{G}_1$ and $\mathcal{G}_2$ be the graphs representing the Decalin and Bicyclopentyl molecules (Figure 2). It was previously shown that $\mathcal{G}_1$ and $\mathcal{G}_2$ are not isomorphic but cannot be distinguished by the 1-WL test (Sato, 2020) . Their *Spectral WL* histograms using $m = 1$ with $t = 1$ and $r = 0$ after the initialization phase are shown in Table 5. Since these histograms are different, *Spectral WL* will determine that these graphs are not isomorphic.

$\square$

## B  Ablation study – Full results

Table 6: Ablation study on the COLLAB results. Highest accuracies per feature size.

| GNN | Feature size | Configuration | Accuracy |
|---|---|---|---|
| SP-GIN | 5 | (-1,1,5,none) | 0.777±0.018 |
| SP-GIN | 10 | (-1,1,10,none) | 0.780±0.021 |
| SP-GIN | 20 | (-1,1,20,max) | 0.788±0.021 |
| SP-GCN | 5 | (-2,2,5,none) | 0.774±0.025 |
| SP-GCN | 10 | (-2,2,10,none) | 0.778±0.019 |
| SP-GCN | 20 | (-2,2,5,MMM) | 0.787±0.021 |
| SP-GAT | 5 | (-1,1,5,none) | 0.758±0.022 |
| SP-GAT | 10 | (-1,1,10,none) | 0.758±0.015 |
| SP-GAT | 20 | (-2,2,20,none) | 0.764±0.016 |
| SP-GraphSAGE | 5 | (-1,1,5,none) | 0.787±0.017 |
| SP-GraphSAGE | 10 | (-1,1,10,none) | 0.795±0.014 |
| SP-GraphSAGE | 20 | (-3,3,10,max) | 0.779±0.015 |

Table 7: Ablation study on the NCI1 results. Highest accuracies per feature size.

| GNN | Features size | Configuration | Accuracy |
|---|---|---|---|
| SP-GIN | 5 | (-1,1,5,none) | 0.806±0.024 |
| SP-GIN | 10 | (-2,2,5,max) | 0.812±0.014 |
| SP-GIN | 20 | (-2,2,10,max) | 0.812±0.014 |
| SP-GCN | 5 | (-1,1,5,none) | 0.801±0.021 |
| SP-GCN | 10 | (-2,2,5,max) | 0.807±0.015 |
| SP-GCN | 20 | (-2,2,20,none) | 0.811±0.014 |
| SP-GAT | 5 | (-2,2,5,none) | 0.806±0.015 |
| SP-GAT | 10 | (-2,2,10,none) | 0.807±0.014 |
| SP-GAT | 20 | (-3,3,10,max) | 0.805±0.014 |
| SP-GraphSAGE | 5 | (-1,1,5,none) | 0.821±0.010 |
| SP-GraphSAGE | 10 | (-2,2,5,max) | 0.816±0.010 |
| SP-GraphSAGE | 20 | (-1,1,10,max) | 0.817±0.018 |

Table 8: Ablation study on COLLAB results on GIN

| GNN | Feature size | Configuration | Accuracy |
|---|---|---|---|
| SP-GIN | 5 | (-2,2,5,none) | 0768±0.015 |
| SP-GIN | 5 | (-1,1,5,none) | 0.777±±0.018 |
| SP-GIN | 10 | (-2,2,5,max) | 0.777±±0.014 |
| SP-GIN | 10 | (-2,2,10,none) | 0.776±±0.017 |
| SP-GIN | 10 | (-1,1,10,none) | 0.780±±0.021 |
| SP-GIN | 10 | (-3,3,10,none) | 0.763±±0.020 |
| SP-GIN | 10 | (-1,1,5,max) | 0.778±±0.022 |
| SP-GIN | 20 | (-2,2,5,'MMM') | 0.755±±0.019 |
| SP-GIN | 20 | (-2,2,10,max) | 0.786±±0.020 |
| SP-GIN | 20 | (-1,1,10,max) | 0.788±±0.021 |
| SP-GIN | 20 | (-3,3,10,max) | 0.772±±0.012 |
| SP-GIN | 20 | (-2,2,20,none) | 0.780±±0.018 |
| SP-GIN | 20 | (-3,3,20,none) | 0.770±±0.015 |

Table 9: Ablation study on COLLAB results on GCN

| GNN | Feature size | Configuration | Accuracy |
|:---:|:---:|:---:|:---:|
| SP-GCN | 5 | (-2,2,5,none) | 0.774±±0.025 |
| SP-GCN | 5 | (-1,1,5,none) | 0.760±±0.016 |
| SP-GCN | 10 | (-2,2,5,max) | 0.777±±0.024 |
| SP-GCN | 10 | (-2,2,10,none) | 0.778±±0.019 |
| SP-GCN | 10 | (-1,1,10,none) | 0.762±±0.022 |
| SP-GCN | 10 | (-3,3,10,none) | 0.777±±0.019 |
| SP-GCN | 10 | (-1,1,5,max) | 0.771±±0.021 |
| SP-GCN | 20 | (-2,2,5,'MMM') | 0.787±±0.021 |
| SP-GCN | 20 | (-2,2,10,max) | 0.774±±0.020 |
| SP-GCN | 20 | (-1,1,10,max) | 0.770±±0.021 |
| SP-GCN | 20 | (-3,3,10,max) | 0.784±±0.023 |
| SP-GCN | 20 | (-2,2,20,none) | 0.770±±0.023 |
| SP-GCN | 20 | (-3,3,20,none) | 0.780±±0.024 |

Table 10: Ablation study on COLLAB results on GAT

| GNN | Feature size | Configuration | Accuracy |
|:---:|:---:|:---:|:---:|
| SP-GAT | 5 | (-2,2,5,none) | 0.753±±0.022 |
| SP-GAT | 5 | (-1,1,5,none) | 0.758±±0.022 |
| SP-GAT | 10 | (-2,2,5,max) | 0.750±±0.024 |
| SP-GAT | 10 | (-2,2,10,none) | 0.753±±0.019 |
| SP-GAT | 10 | (-1,1,10,none) | 0.758±±0.015 |
| SP-GAT | 10 | (-3,3,10,none) | 0.750±±0.020 |
| SP-GAT | 10 | (-1,1,5,max) | 0.758±±0.022 |
| SP-GAT | 20 | (-2,2,5,'MMM') | 0.749±±0.021 |
| SP-GAT | 20 | (-2,2,10,max) | 0.763±±0.022 |
| SP-GAT | 20 | (-1,1,10,max) | 0.759±±0.018 |
| SP-GAT | 20 | (-3,3,10,max) | 0.758±±0.020 |
| SP-GAT | 20 | (-2,2,20,none) | 0.764±±0.016 |
| SP-GAT | 20 | (-3,3,20,none) | 0.758±±0.022 |

Table 11: Ablation study on COLLAB results on GraphSAGE

| GNN | Feature size | Configuration | Accuracy |
|---|---|---|---|
| SP-GraphSAGE | 5 | (-2,2,5,none) | 0.779±±0.018 |
| SP-GraphSAGE | 5 | (-1,1,5,none) | 0.787±±0.017 |
| SP-GraphSAGE | 10 | (-2,2,5,max) | 0.778±±0.020 |
| SP-GraphSAGE | 10 | (-2,2,10,none) | 0.779±±0.016 |
| SP-GraphSAGE | 10 | (-1,1,10,none) | 0.795±±0.014 |
| SP-GraphSAGE | 10 | (-3,3,10,none) | 0.778±±0.024 |
| SP-GraphSAGE | 10 | (-1,1,5,max) | 0.792±±0.019 |
| SP-GraphSAGE | 20 | (-2,2,5,'MMM') | 0.776±±0.021 |
| SP-GraphSAGE | 20 | (-2,2,10,max) | 0.774±±0.019 |
| SP-GraphSAGE | 20 | (-1,1,10,max) | 0.773±±0.025 |
| SP-GraphSAGE | 20 | (-3,3,10,max) | 0.779±±0.015 |
| SP-GraphSAGE | 20 | (-2,2,20,none) | 0.778±±0.020 |
| SP-GraphSAGE | 20 | (-3,3,20,none) | 0.774±±0.022 |

Table 12: Ablation study on NCI1 results on GIN

| GNN | Feature size | Configuration | Accuracy |
|---|---|---|---|
| SP-GIN | 5 | (-2,2,5,none) | 0.806±±0.024 |
| SP-GIN | 5 | (-1,1,5,none) | 0.809±±0.014 |
| SP-GIN | 10 | (-2,2,5,max) | 0.812±±0.014 |
| SP-GIN | 10 | (-2,2,10,none) | 0.803±±0.019 |
| SP-GIN | 10 | (-1,1,10,none) | 0.803±±0.017 |
| SP-GIN | 10 | (-3,3,10,none) | 0.808±±0.016 |
| SP-GIN | 10 | (-1,1,5,max) | 0.804±±0.015 |
| SP-GIN | 20 | (-2,2,5,'MMM') | 0.809±±0.018 |
| SP-GIN | 20 | (-2,2,10,max) | 0.812±±0.014 |
| SP-GIN | 20 | (-1,1,10,max) | 0.810±±0.016 |
| SP-GIN | 20 | (-3,3,10,max) | 0.807±±0.015 |
| SP-GIN | 20 | (-2,2,20,none) | 0.808±±0.021 |
| SP-GIN | 20 | (-3,3,20,none) | 0.811±±0.016 |

Table 13: Ablation study on NCI1 results on GCN

| GNN | Feature size | Configuration | Accuracy |
|:---:|:---:|:---:|:---:|
| SP-GCN | 5 | (-2,2,5,none) | 0.801±±0.019 |
| SP-GCN | 5 | (-1,1,5,none) | 0.801±±0.021 |
| SP-GCN | 10 | (-2,2,5,max) | 0.807±±0.015 |
| SP-GCN | 10 | (-2,2,10,none) | 0.804±±0.010 |
| SP-GCN | 10 | (-1,1,10,none) | 0.803±±0.016 |
| SP-GCN | 10 | (-3,3,10,none) | 0.795±±0.016 |
| SP-GCN | 10 | (-1,1,5,max) | 0.806±±0.021 |
| SP-GCN | 20 | (-2,2,5,'MMM') | 0.810±±0.014 |
| SP-GCN | 20 | (-2,2,10,max) | 0.805±±0.011 |
| SP-GCN | 20 | (-1,1,10,max) | 0.803±±0.018 |
| SP-GCN | 20 | (-3,3,10,max) | 0.805±±0.017 |
| SP-GCN | 20 | (-2,2,20,none) | 0.811±±0.014 |
| SP-GCN | 20 | (-3,3,20,none) | 0.810±±0.016 |

Table 14: Ablation study on NCI1 results on GAT

| GNN | Feature size | Configuration | Accuracy |
|---|---|---|---|
| SP-GAT | 5 | (-2,2,5,none) | 0.806±±0.015 |
| SP-GAT | 5 | (-1,1,5,none) | 0.801±±0.018 |
| SP-GAT | 10 | (-2,2,5,max) | 0.806±±0.017 |
| SP-GAT | 10 | (-2,2,10,none) | 0.807±±0.014 |
| SP-GAT | 10 | (-1,1,10,none) | 0.790±±0.016 |
| SP-GAT | 10 | (-3,3,10,none) | 0.806±±0.021 |
| SP-GAT | 10 | (-1,1,5,max) | 0.800±±0.018 |
| SP-GAT | 20 | (-2,2,5,'MMM') | 0.791±±0.018 |
| SP-GAT | 20 | (-2,2,10,max) | 0.792±±0.015 |
| SP-GAT | 20 | (-1,1,10,max) | 0.794±±0.021 |
| SP-GAT | 20 | (-3,3,10,max) | 0.805±±0.014 |
| SP-GAT | 20 | (-2,2,20,none) | 0.798±±0.013 |
| SP-GAT | 20 | (-3,3,20,none) | 0.798±±0.013 |

Table 15: Ablation study on NCI1 results on GraphSAGE

| GNN | Feature size | Configuration | Accuracy |
|---|---|---|---|
| SP-GraphSAGE | 5 | (-2,2,5,none) | 0.812±±0.010 |
| SP-GraphSAGE | 5 | (-1,1,5,none) | 0.821±±0.011 |
| SP-GraphSAGE | 10 | (-2,2,5,max) | 0.816±±0.010 |
| SP-GraphSAGE | 10 | (-2,2,10,none) | 0.810±±0.021 |
| SP-GraphSAGE | 10 | (-1,1,10,none) | 0.810±±0.010 |
| SP-GraphSAGE | 10 | (-3,3,10,none) | 0.811±±0.009 |
| SP-GraphSAGE | 10 | (-1,1,5,max) | 0.810±±0.013 |
| SP-GraphSAGE | 20 | (-2,2,5,'MMM') | 0.807±±0.010 |
| SP-GraphSAGE | 20 | (-2,2,10,max) | 0.809±±0.011 |
| SP-GraphSAGE | 20 | (-1,1,10,max) | 0.817±±0.018 |
| SP-GraphSAGE | 20 | (-3,3,10,max) | 0.811±±0.013 |
| SP-GraphSAGE | 20 | (-2,2,20,none) | 0.815±±0.018 |
| SP-GraphSAGE | 20 | (-3,3,20,none) | 0.816±±0.013 |

## C  Addition experiments settings

Table 16: Graph classification datasets statistics

| Name | Graphs | Classes | Avg. Nodes | Avg. Edges |
|---|---|---|---|---|
| MUTAG | 188 | 2 | 17.93 | 19.79 |
| PTC | 336 | 2 | 13.97 | 14.32 |
| PROTEINS | 1113 | 2 | 39.06 | 72.82 |
| NCI1 | 4110 | 2 | 29.87 | 32.30 |
| COLLAB | 5000 | 3 | 74.49 | 2457.78 |
| IMDB-BINARY | 1000 | 2 | 19.77 | 96.53 |
| IMDB-MULTI | 1500 | 3 | 13.00 | 65.94 |
| REDDIT-BINARY | 2000 | 2 | 429.63 | 497.75 |
| REDDIT-MULTI | 4999 | 5 | 508.52 | 594.87 |

Table 17: Graph classification datasets statistics

| Name | Classes | Avg. Nodes | Avg. Edges |
|---|---|---|---|
| CORA | 7 | 2,708 | 5,429 |
| CITESEER | 6 | 3,327 | 4,732 |
| PUBMED | 3 | 19,717 | 44,338 |
| PPI | 121 (multilabel) | 56944 | 818716 |

We used a single GeForce RTX™ 3090 to run each benchmark. The time took to train and test each GNN on a single benchmark was approximately one to four hours, depends on the benchmark size.

# D Approximation analysis of the scaling technique

The scaling technique can be theoretically analyzed using the following theorems:

**Theorem 6.** *The upper bound for the approximation error of the heat kernel using implicit Euler is proportional to the inverse of the number of iterations, i.e., $error = O(\frac{1}{num\_iterations})$.*

*Proof.* The step size in each iteration is proportional to the upper bound of the approximation error (Larsson & Thomée, 2003). Since the number of iterations is inversely proportionate to the step size, we get that the approximation error is proportionate to the inverse of the number of iterations. □

**Theorem 7.** *The upper bound for the approximation error of the heat kernel using truncated spectrum decays exponentially with respect to the value of the smallest eigenvalue truncated, i.e., $error = O(e^{-\lambda_{k+1}t})$.*

*Proof.*

$$error = ||\Sigma_{i=k+1}^{n} e^{-\lambda_i t} \phi_i \phi_i^T|| \leq (n-k)e^{-\lambda_{k+1}t} \tag{10}$$

□

We also conducted an empirical approximation analysis on the scaling technique. For that, we used graphs from the REDDIT-BINARY dataset. statistics regarding REDDIT-BINARY are available in the appendix. We chose two points in time, $t_1 = 0.1$ and $t_2 = 1$, and computed their approximated heat kernel matrices of the graphs. For $t_1$ we used explicit Euler iterations since $t_1$ is relatively small. For $t_2$ we truncated the largest eigenvalues since $t_2$ is relatively large.

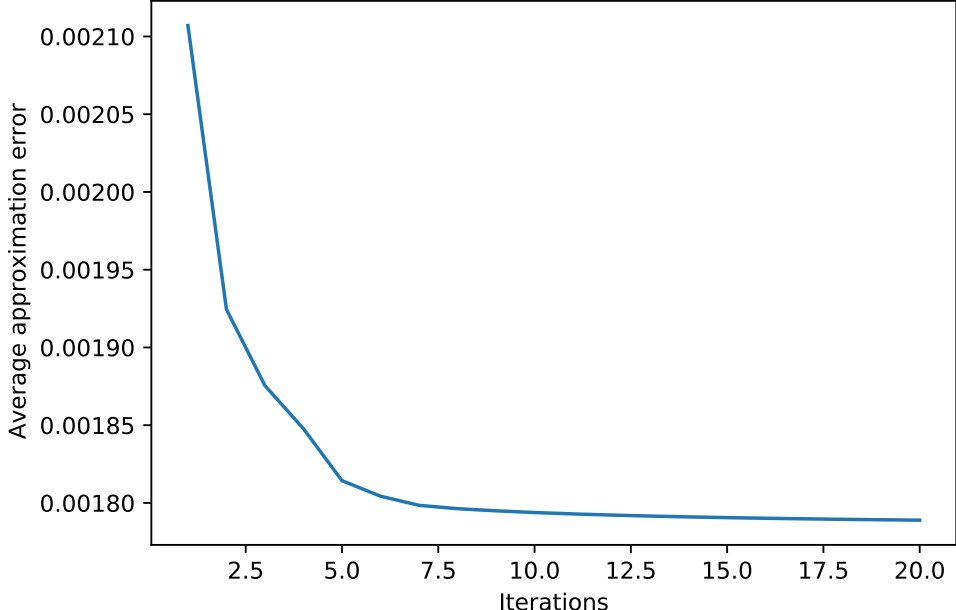

Figure 6: Average approximation error by the number of iterations.

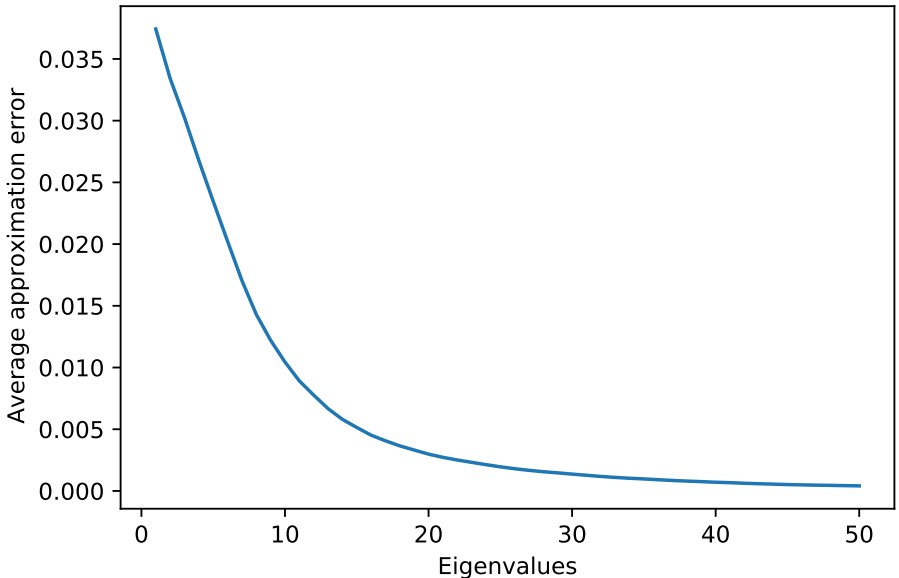

Figure 7: Average approximation error by the number of eigenvalues.

For both $t_1$ and $t_2$ the average approximation error decays exponentially with the number of Euler iterations and the number of eigenvalues respectively. For $t < 0.1$ or $t > 10$ the approximation error will decay faster. From these results, we can understand that the suggested method of approximation is able to reduce the computation time of the heat kernel matrix dramatically, and our suggested spectral per-coloring specifically, while still being able to approximate it accurately.

