# OpenReview forum: "Weisfeiler and Leman Go Infinite: Spectral and Combinatorial Pre-Colorings"
_TMLR — Accepted by TMLR_

### Review · Reviewer_fyxi · 2022-12-05

**Summary Of Contributions:**

This paper describes methods of improving the expressiveness of MPNNs beyond the 1-WL graph isomorphism test. It does so by preprocessing the graph using a coloring defined by the eigendecomposition of the graph's Laplacian. This is equivariant to vertex permutations, and provably allows the GNN to distinguish between graphs in a finer way. Experiments are performed on synthetic and real-world datasets showing that classification accuracy increases when these labelings are added and for which settings they are most effective.

**Audience:**

Yes

**Claims And Evidence:**

Yes

**Requested Changes:**

Critical points:

- The proof of Theorem 2 is hard to follow; I think it needs more details and clearer writing. Additionally, it only proves the statement for $k = 3$.

Minor points:

- In table 3: What does "specific" mean in the numbers?

- The $k$-WL test is introduced but never precisely defined in the text (only the 1-WL test is fully defined). This section in the preliminaries could use a little bit more exposition.

- There are some formatting errors in the document. In the proof of Theorem 1, it says "wide, labelwidth=!, labelindent=0pt".

- On page 8, "provingly" should be replaced with "provably".

- For the base case of the proof of Theorem 1, "equivarinat" is a typo.

- Instead of k-WL, it should say $k$-WL; there are several other parts where the correct math font is not used.

**Strengths And Weaknesses:**

Strengths

- The colorings defined seem to provide a genuine improvement in GNN performance beyond the constant feature representation.
- The experiments are comprehensive and examine a large variety of labeling configurations and GNN architectures.
- The claims are theoretically well-justified. There is a detailed discussion comparing the expressiveness of the WL hierarchy under different initial colorings.

Weaknesses

- It's not clear to me why we need to compute the heat kernel at various times. Is it not just better to encode the eigenvectors themselves? This could benefit from a more intuitive explanation.

- Spectral labelings are compared with constant features, but not with random node features. If GNNs can reap the same benefit in expressivity from random node features, I do not see a reason to use the computationally more expensive spectral colorings.

- The writing could use some improvement (see Requested Changes).

---

> ### Author Response · Authors · 2022-12-07
> **Comment for Reviewer fyxi**
>
> First and foremost, we thank you for the time and effort you put into your review and for your helpful feedback and remarks. We are glad that you found that our method provides a genuine improvement in GNNs, the experiments are comprehensive, and the claims in our paper are theoretically well-justified, as you mentioned in the Strengths section.
>
> Regarding the weaknesses of the paper as you mentioned in the Weaknesses section:
> 1. In the paragraph between Theorem 4 and Theorem 5, and at the end of Section 5 we explain and demonstrate (using a relatively simple setting experiment) the advantage of using the heat kernel instead of using the eigenvectors themselves. In a nutshell, a straightforward node coloring that is based on the eigendecomposition of the graph Laplacian is not a permutation equivariant. This is because a single Laplacian matrix may have multiple different eigendecompositions.
>
> 2. Random features are not permutation equivariant. At the end of Section 5, we explain and demonstrate the advantage of using the heat kernel instead of using the random features. Moreover, we compared our suggested technique with previously suggested permutation equivariant pre-colorings (LDP and 3-SI) and outperformed them on almost every benchmark.
>
> 3. We will fix all requested changes.

---

> > ### Comment · Reviewer_fyxi · 2023-01-04
> > **Response to comment**
> >
> > Thank you for your response and for addressing my concerns. Indeed, your method has advantages over using a coloring based on a Laplacian eigendecomposition or random features. I see that your submission still has some of the typos I mentioned above. I look forward to seeing the revised version.

---

### Review · Reviewer_M4aF · 2022-12-28

**Summary Of Contributions:**

This paper studies the WL test for message passing of GNNs. It proposes a new message passing using the spectral information in heat kernel on the graph, and the WL algorithm. It shows in theory the spectral WL is more expressive than 1-WL which is the expressiveness of many GNN methods. A scaling method for reducing the computational cost of spectral message passing is discussed. Experiments for the proposed spectral method for message passing on synthetic and real datasets for both node-level and graph-level classifications are carried on and compared with existing GNN models.

**Audience:**

Yes

**Claims And Evidence:**

Yes

**Requested Changes:**

1. The spectral WL which is the focus of the paper appears in Section 4. It can be moved forward.

2. The discussion of \Detal(k-WL) is not major topic in the paper. It can be moved to the appendix.

3. There are some typos to be corrected. For example, on page 4, in Theorem 4 “are equal ” is a repetition of $\equiv$.

4. In Table 1, at least 1 decimal digit should be preserved when showing accuracy like Table 2. The variance of the accuracy is big.


**Strengths And Weaknesses:**

**Strengths:**

1. Graph heat kernel provides a new spectral method for designing message passing.

2. The proposed spectral message passing is implemented by the spectral WL which expressivity is easily measured by the WL algorithm.

3. Proved the equivariance of \Delta(k-WL) and Ck-WL.

**Weakness:**

1. For large graphs, the computational cost of the spectral method may be high.

2. The error of truncation approximation in spectral WL is not discussed.

3. The expressiveness of \Delta(k-WL) is discussed in theory. However, it is not used in studying the proposed spectral WL which is the focus of the paper.

4. In experiments, most SOTA results are achieved by the spectral WL-pretrained GNNs, which is not sufficient to tell the superiority of the proposed message passing.

5. There are other spectral GNNs, like Graph Wavelet Neural Network [Wu et al. (ICLR, 2018)], and Framelet convolution [Zheng et al. (ICML, 2021)]. These need to be compared with the spectral WL of the paper.

6. If one uses spectral WL as a message-passing module of GNNs, can the GNNs be very deep? That is, will spectral WL limit oversmoothing?

---

> ### Author Response · Authors · 2022-12-30
> **Comment for Reviewer M4aF**
>
> Dear reviewer M4aF,
>
> We are  grateful for your thorough review and appreciate the effort you have put in to provide constructive feedback on our work.  Regarding the notations you mentioned in the strength section;  in our paper, we do not aim to “provide a new spectral method for designing message passing” or to “propose spectral message passing”. We aim to suggest a new spectral pre-coloring, which is a spectral-based approach to generate node features as a pre-process before the learning process (message passing) of the GNNs begins. We prove that using this pre-coloring on the relatively simple but very efficient Message Passing Neural Networks enables them to break the limit in expressivity in theory, and improve their performance in practice.
>
> Regarding the weaknesses of the paper as you mentioned in the Weaknesses section:
>
> 1. In the last paragraphs of section 4 we discuss in detail how one can use model order reduction to significantly reduce  the computation time required for the spectral pre-coloring. In addition, even though we used benchmarks of graphs with hundreds and thousands of nodes, we were able to compute the standard (not truncated) spectral pre-coloring in a reasonable time (no longer than a few minutes for graphs with ~10K nodes, on Intel(R) Xeon(R) CPU 2.20GHz ).
>
> 2. We will add an analysis of the approximation error.
>
> 3. As mentioned in the introduction section in our paper, the expressiveness of \Delta(k-WL) serves as motivation for the two phase approach in general. Moreover, we use Theorem 1 (presented in the  expressiveness of \Delta(k-WL) section) to prove our claim about the expressive power of the spectral pre-coloring.
>
> 4. We did not use any pre-trained networks in the paper. The spectral pre-coloring and the other methods we compared to does not have learnable parameters and hence does not require any training prior to their use. We would appreciate it if you could pinpoint the details for determining that the experiments were utilized on pre-trained networks so that we can enhance the clarity of our paper.
>
> 5. As mentioned earlier, the purpose of our paper is to suggest a novel pre-coloring to improve the performance of existing MPNN architectures, and not to build new GNNs like Wavelet Neural Network or Framelet convolution. In the experiments section we compared our method to previously suggested equivariant pre-colorings, and outperformed them on the benchmarks. We also demonstrated the superiority of our suggested method compared to other non-equivariant pre-coloring techniques, both spectral based and random based.
>
> 6. Good point! From our results on the node classification it could be possible, but it requires further research. We will add this notion in the discussion section of the final version of our paper.
>
> Regarding the requested changes:
>
> 1+2.  We strongly believe that since the discussion of \Detal(k-WL) serves as a theoretical background and justification for the use of pre-colorings in general, and also contains Theorem 1 which is later being used to prove the expressive power of the spectral WL, it should appear in the main part of the paper, before the spectral WL section.
>
> 3+4. Thanks for the issues you found, we will fix both.
>
> Given the statements we previously made, we request your response before we make any revisions.

---

> > ### Author Response · Authors · 2023-01-02
> > **Comment**
> >
> > Due to the short period of rebuttal and revision, until we receive your response,  we will only add the changes we declared in our comment above.

---

> > > ### Author Response · Authors · 2023-01-09
> > > **Note for Reviewer M4aF**
> > >
> > > We didn't find any "are equal" in Theorem 4. If you meant to refer to the "are equal" in Theorem 2, then it is not a typo.

---

> > ### Comment · Reviewer_M4aF · 2023-02-06
> > **about revisions**
> >
> > Thanks for the responses.
> > 2. if you can add theorems/proofs about approximation error, it would be great to see it in the revision.
> > Other reply are good.

---

> > > ### Author Response · Authors · 2023-02-06
> > > **comment about revisions**
> > >
> > > We are happy that you found our replies good.
> > > We will soon add theorems regarding the approximation error to the revised version of our paper.

---

> > > > ### Author Response · Authors · 2023-02-09
> > > > **Revision**
> > > >
> > > > We have updated the revised version of the paper.
> > > > You can find the theorems in the appendix.

---

### Review · Reviewer_6e9B · 2023-01-01

**Summary Of Contributions:**

The paper studies expressivity of message passing GNNs with certain pre-colorings of the input nodes, in particular based on spectral features.
The authors study improvements in the expressivity of 1-WL approaches (related to basic GNNs) with different pre-colorings, including colorings given by k-WL procedures, as well as spectral features obtained from a few steps of graph diffusions.
These pre-colorings are then tested empirically on various architectures and datasets, where it is found that they consistently improve performance across a range of tasks.

**Audience:**

Yes

**Claims And Evidence:**

Yes

**Requested Changes:**

See above for general questions that could improve the paper.

Other minor comments:
- many references should be fixed (citet -> citep)
- the figures and tables also seem quite big, could you make them smaller?
- p.3, MPGNN paragraph:
   * not sure what "descriptor theoretical" means.
   * please include references for the results mentioned at the end of the paragraph ("it was proved", "it has been shown")
- theorem 3: K -> k

**Strengths And Weaknesses:**

Strengths: the paper is well written, and provides good empirical evidence that including "colorings" as additional features in the input nodes can improve performance of simple GNN architectures across many different tasks.

Weaknesses/general comments:

* The expressiveness results are mostly about comparing to the relatively weak class of 1-WL colorings. It would be good to have a more crisp characterization of what class of graphs spectral colorings can further discriminate, e.g. can they beat k-WL for k > 1?

* If I understand correctly, the "infinite"/"ad infinitum" mentioned in the title/abstract refers to iteratively refining colorings. While it is clear that this can improve expressivity, can this improvement be quantified somehow? e.g. in Thm 3 you show "strict" improvements in expressivity with k-WL colorings, but is there a way to quantify how much the corresponding class of graphs grows?

* it seems that the spectral colorings considered in the paper could be obtained with a few spectral filters with a one-layer spectral GNN (e.g. the diffusion can be implemented with exponential filters). If so, what are benefits of using pre-colorings instead of using such richer spectral architectures? A more detailed comparison would be useful here.

* Recent works have compared the use of richer graph-based input features to using GNNs, see, e.g. this paper: https://arxiv.org/abs/2010.15116 and references therein. How does the conclusions of the present manuscript relate to this line of work?

---

> ### Author Response · Authors · 2023-01-02
> **Comment for Reviewer 6e9B**
>
> Dear reviewer 6e9B,
>
> Thank you very much for taking the time to review our manuscript and for providing such valuable and constructive feedback. We appreciate the effort you have put into carefully reading and evaluating our work.
> We are glad that you found our method good for “ improving  performance of simple GNN architectures”, and that the work is ”well written, and provides good empirical evidence”, as you mentioned in the strength section.
>
>
> Regarding the general comments you raised:
>
> 1. The exact expressive power of spectral approaches is currently unknown and it is stated as an open question in ^1. However, in the same paper, the author showed that spectral methods based on the adjacency matrix are bounded in their expressive power by 2-WL. The authors in ^2  reached the same conclusion for the graph Laplacian and other matrices that are built from the adjacency matrix.  We will add this notion to the final version of our paper.
>
>
> 2. The exact characterization of the WL hierarchy is a topic of ongoing research in the literature. In our paper, we cited several works that proved the expressivity of specific and general K’s over specific classes of graphs. Other works, such as ^3 and ^4, try to refine the WL hierarchy and define intermediate levels of expressivity between any k and k+1. Finding the exact characterization of the WL hierarchy requires further search which is beyond the scope of our work.
>
>
> 3. We will list here the advantages (or differences) of using our spectral pre-coloring over using spectral GNNs:
>
> a. In contrast to standard spectral GNNs, our approach does not require learning, and hence it is only needed to be computed once though the whole learning process. This property enables our technique to be used as a simple off-the-shelf tool for GNNs in general.
>
> b.  As demonstrated in our paper (the paragraph between Theorem 4 and Theorem 5), many spectral GNNs depend on the eigendecomposition representation. This dependency makes the GNNs non-equivariant, which is an undeniable property. Our suggested spectral pre-coloring is permutation equivariant.
>
> c. Many standard spectral GNNs suffer from unscalability ^5 and there is ongoing research regarding making them more efficient. Not only that our technique only needed to be computed once per graph during the learning process, but in the paper we also suggest an approximation for our method that reduces the computation time dramatically.
>
> We hope that you find this answer useful.
>
> 4. Thank you for enlightening us with this interesting line of works. In the provided paper, the authors mostly support the use of GNNs upon using GA-MLP (basically a pre-coloring combined with an MLP):
>
> a. “Finding graph pairs that several GA-MLPs cannot distinguish while GNNs can”.
>
> b. “can be approximated by GNNs but not GA-MLPs both in theory and numerically”.
>
> c. “demonstrate the limitations of GA-MLPs”.
>
> In our paper, on the other hand, we proved how pre-coloring can improve the expressivity of GNNs ad
> infinitum in theory and gave a concrete instance of spectral pre-coloring that improves GNNs both in
> theory and in practice. We believe that these two approaches can co-exist when applying pre-colorings
>  directly on GNNs instead of MLPs, as implemented in our paper. Thus one can enjoy the best of both worlds.
>
> Thank you for finding the typos, the unclear points, and other issues in our paper. We will fix them all in the revised version.
>
> We hope that we managed to answer all of your concerns. In case you still have unresolved concerns, please let us know.
>
>
>
> ^1“On the power of combinatorial and spectral invariants”, Furer, 2010.
>
> ^2”Weisfeiler–Leman, Graph Spectra, and Random Walks”, Rattan & Seppelt, 2021.
>
> ^3”A Practical, Progressively-Expressive GNN”, Zhao et al., 2022.
>
> ^4”SpeqNets: Sparsity-aware Permutation-equivariant Graph Networks”, Morris et al., 2022.
>
> ^5”ChebNet: Efficient and Stable Constructions of Deep Neural Networks with Rectified Power Units using Chebyshev Approximations”, Tang et al.,  2019.

---

> > ### Author Response · Authors · 2023-01-09
> > **Note for Reviewer 6e9B**
> >
> > We didn't find any "descriptor theoretical" in the paper. If you meant to refer to the " theoretical superiority" in the MPGNN paragraph, the meaning of this phrase is that it has been shown that MPGNNs based on summation have higher discriminative power than MPGNNS based on other aggregation operations.

---

### Author Response · Authors · 2023-01-09
**Revised version**

Dear action editor and reviewers,

We want to thank again for the reviewers for all their constructive comments and helpful feedback.
We uploaded a revised version of our paper and marked most of the changes in RED for convenience.
Here is the full list of changes we added to the paper:

1.We rewrote the proof of Theorem 2.

2.We fixed the “specific” issue in table 3 and table 2.

3.We added a reference for the definition of k-WL in the preliminaries.

4.We fixed formatting errors in the proof of Theorem 1.

5.We replaced  "provingly" with  "provably" right after Theorem 3.

6.We fix the "equivarinat" typo.

7.We switched any  k-WL to  k(italic)-WL.

8.We added decimal digits in table 1.

9.We added approximation analysis of the scaling technique to the appendix. We also added the full experiment to the Git repository.

10.We added the notion of possible support to the overquashing problem in the discussion section.

11.We fixed the relevant references (citet -> citep).

12.We reduced the sizes of tables and figures where the reduction did not harm the quality of the presentation.

13.We included the relevant references in the MPGNN paragraph.

---

### Decision · Action_Editors · 2023-03-25

**Recommendation:** Accept with minor revision

**Comment:**

In Section 3 it is mentioned that the expressive power of 1-WL can be improved to anywhere in the k-WL hierarchy when using precoloring. If this is to be interpreted as stating that 1-WL with appropriate precoloring can be made as powerful as any k-WL, it would be good to add an explicit statement to this amount, otherwise rephrase for clarify.

Section 4 presents a theorem stating that spectral precoloring is strictly more expressive than 1-WL. It would be good to mention how the proposed spectral precoloring compares, or how it is known to compare, with diagonal k-WL precoloring and how 1-WL with the proposed precoloring compares with k-WL.

The article still has a number of typos, including the following.
* There are repeated references
* There are typos in the citation formatting. For orientation see section References in https://www.jmlr.org/format/format.html.
* pg 3 k≥2 vs $k\geq2$; pg 5 k-1 vs $k-1$, k-WL vs $k$-WL
* pg 3, pg 4 \Delta(k - WL) vs \Delta(\text{$k$-WL})
* pg 4 t vs $t$
* pg 4 i-th vs $i$-th
* pg 5 space in $1$-$\Delta(...)$
* pg 5, pg 6 misplaced . before reference
* Tables 1, 2, 3 reference from the text seems to be missing. Font size in the tables seems to be inconsistent.

**Audience:**

The topic is of substantial interest in TMLR's audience.

**Claims And Evidence:**

The article investigates precoloring to improve expressivity of MPGNNs. It presents theorems to show that precoloring can improve expressivity and proposes a particular spectral precoloring. The proposed method is evaluated on diverse experiments. The claims are sufficiently well supported by theorems and experiments, except for a few requests given below.